# Manipulation Direction: Evaluating Text-Guided Image Manipulation Based on Similarity between Changes in Image and Text Modalities

**DOI:** 10.3390/s23229287

**Published:** 2023-11-20

**Authors:** Yuto Watanabe, Ren Togo, Keisuke Maeda, Takahiro Ogawa, Miki Haseyama

**Affiliations:** 1Graduate School of Information Science and Technology, Hokkaido University, N-14, W-9, Kita-ku, Sapporo 060-0814, Hokkaido, Japan; y_watanabe@lmd.ist.hokudai.ac.jp; 2Faculty of Information Science and Technology, Hokkaido University, N-14, W-9, Kita-ku, Sapporo 060-0814, Hokkaido, Japan; togo@lmd.ist.hokudai.ac.jp (R.T.); maeda@lmd.ist.hokudai.ac.jp (K.M.); ogawa@lmd.ist.hokudai.ac.jp (T.O.)

**Keywords:** text-guided image manipulation, generative adversarial network, evaluation metric, manipulation direction

## Abstract

At present, text-guided image manipulation is a notable subject of study in the vision and language field. Given an image and text as inputs, these methods aim to manipulate the image according to the text, while preserving text-irrelevant regions. Although there has been extensive research to improve the versatility and performance of text-guided image manipulation, research on its performance evaluation is inadequate. This study proposes Manipulation Direction (MD), a logical and robust metric, which evaluates the performance of text-guided image manipulation by focusing on changes between image and text modalities. Specifically, we define MD as the consistency of changes between images and texts occurring before and after manipulation. By using MD to evaluate the performance of text-guided image manipulation, we can comprehensively evaluate how an image has changed before and after the image manipulation and whether this change agrees with the text. Extensive experiments on Multi-Modal-CelebA-HQ and Caltech-UCSD Birds confirmed that there was an impressive correlation between our calculated MD scores and subjective scores for the manipulated images compared to the existing metrics.

## 1. Introduction

Research fields of computer vision and natural language processing have gradually developed by interacting with each other. Particularly, with the emergence of recent deep learning techniques, such as a convolutional neural network, long short-term memory [1], transformer [2], and vision transformer [3], the compatibility of these research fields has considerably enhanced. This trend has prompted proposals related to new tasks, such as visual question answering (VQA) [4,5], cross-modal image–text retrieval [6,7], image captioning [8,9], referring image segmentation [10,11], and text-to-image synthesis [12,13].

With the emergence of generative models such as the variational autoencoder [14], generative adversarial network [15], and a diffusion model [16], the research related to text-to-image synthesis has become more active. Inspired by the idea of using text prompts to control generated images, several studies [17,18,19,20,21,22,23,24,25] have proposed text-guided image manipulation methods that manipulate images using natural language. Text-guided image manipulation could improve the flexibility of traditional image manipulation methods, such as image colorization [26,27], image inpainting [28,29], and style transfer [30,31]. These traditional methods focus primarily on specific tasks and can transform only one attribute. Such transformations have limited ability to reflect the user’s demands for image manipulation. Moreover, as shown in Figure 1, text-guided image manipulation methods take an image and text prompt describing where and how to manipulate the image from a user and generates a manipulated image. These methods typically aim to manipulate a text-relevant region according to a text prompt, while preserving text-irrelevant regions. Since the user’s demands are reflected in the manipulated image, such methods achieve user-friendly image manipulation.

Text-guided image manipulation methods reflect great potential, and the establishment of evaluation metrics is necessary for their further development. Since the results of text-guided image manipulation are based on a text given by the user, subjective evaluations are the most reliable, but the collection of subjective evaluations requires tremendous time and effort. Considering this, several metrics for automatically evaluating image manipulation performance have been proposed and practically implemented [18,33,34,35]. First, to assess the quality and aesthetics of a manipulated image, existing studies applied structural similarity (SSIM) [33], inception score (IS) [34], and Fréchet inception distance (FID) [35]. Furthermore, to automatically evaluate the performance, considering image manipulation accuracy concerning correspondence with a text prompt and the preservation of text-irrelevant regions is necessary. Image manipulation evaluation relative to a text prompt is performed by measuring the similarity between the text prompt and the manipulated image (e.g., cosine similarity). Meanwhile, the evaluation of the preservation of text-irrelevant regions is performed by measuring the similarity between input and manipulated images (e.g., L1 and L2 distances). Such similarities are usually calculated and evaluated separately. Moreover, manipulative precision (MP) [18] uses “cosine similarity between a manipulated image and text prompt” and “L1 distance between the input and manipulated images” and evaluates the accuracy of image manipulation using the text prompt.

Although a research field on image quality assessment has been established, there is insufficient research on evaluation metrics for image manipulation performance. The existing evaluation metrics have limitations in directly evaluating image manipulation according to text prompts. Specifically, as shown in Figure 2, the existing metrics have the following three problems:The similarity between a text prompt and the manipulated image completely ignores how an input image has been changed.The similarity between the input and manipulated images completely ignores regions manipulated according to the text prompt and evaluates the image reconstruction.There are limitations in deriving the performance of image manipulation models from both scores listed above because they commonly have a trade-off relationship.

Specifically, point (a) shows the problem of evaluating only image synthesis from the text prompt, and there are no constraints on the changes made to the input image. Point (b) shows the problem of similarity between the input and manipulated images not evaluating the performance of preserving the text-irrelevant regions only, and instead, evaluating the performance of simple image reconstruction. In point (c), we show that image manipulation causes a trade-off relationship between the similarities shown in points (a) and (b), and it is difficult to derive the accuracy of text-guided image manipulation from them. This indirect evaluation is limited in its ability to collaboratively evaluate how an image has changed before and after its manipulation and whether the change is consistent with the text prompts. If they could be collectively evaluated on a single scale, no trade-off relationship would arise, although this is yet to be resolved. Consequently, previous studies have performed time-consuming and laborious subject experiments and shown numerous samples to supplement this incompleteness.

To address the above problem of indirectly evaluating image manipulation, as shown above, we establish a logical yet simple metric, called Manipulation Direction (MD), for evaluating the accuracy of text-guided image manipulation in this study. We name the evaluation metric “MD” because it is designed to focus on the direction of changes in images and texts that occur before and after image manipulation in the common embedding space. Using MD as an evaluation metric, we can calculate evaluation scores that focus on the similarity between changes in a text provided by a user and changes from the input to the manipulated image. In text-guided image manipulation, a user specifies the desired image manipulation by performing word replacement in the text prompt accompanying an input image. This replacement in the text space can be interpreted as the direction of the image manipulation desired by the user in the image space. When image manipulation is not according to a text prompt, or when text-irrelevant regions are manipulated, the similarity with the direction of the text change inevitably decreases. Moreover, the more similar the direction of changes in the text prompt and the image are, i.e., the higher the MD score, the more accurate we can consider the image manipulation to be. MD comprehensively evaluates the image manipulation according to a text prompt and the preservation of text-irrelevant regions based on the direction of the text change; thus, there is no trade-off relationship, unlike existing evaluation metrics. MD precisely captures the changes between each modality and can directly evaluate the performance of image manipulation according to the text prompts. The eventual purpose of MD is to provide quantitative evaluations of text-guided image manipulation and to calculate evaluation scores that correlate with subjective evaluations of manipulated images. Our primary contributions are summarized as follows.

We newly propose MD, which is an evaluation metric for text-guided image manipulation.MD is defined as the consistency of changes between images and texts occurring before and after manipulation in the common embedding space.MD comprehensively evaluates the image manipulation according to a text prompt and the preservation of text-irrelevant regions on a single scale.

The remainder of this article is organized as follows. In Section 2, we introduce related works on text-guided image manipulation and evaluation approaches for them. In Section 3, we describe our new metric, MD, for evaluating the accuracy of text-guided image manipulation. We present extensive experimental results to verify the effectiveness of MD in Section 4. Finally, we conclude our work in Section 5. Notably, this article is an extension of our previous work [36].

## 2. Related Works

### 2.1. Text-Guided Image Manipulation

Recently, vision-language tasks and text-guided image manipulation, in particular, have gained considerable attention. The major advantage of text-guided image manipulation is that it provides highly flexible image manipulation by receiving text prompts from a user. Text-guided image manipulation methods generally aim to manipulate text-relevant regions according to a text prompt and preserve text-irrelevant regions. Dong et al. [37] first proposed an encoder–decoder architecture that could generate manipulated images that were conditioned on images and text embeddings. Subsequently, Nam et al. [17] proposed a new approach to disentangling fine-grained visual attributes in a text using a text-adaptive discriminator and they succeeded in obtaining a generator that could produce particular visual attributes. These two studies [17,37] focused mainly on manipulating the colors and textures of birds or flowers in the images contained in datasets such as Caltech-UCSD Birds [32] and Oxford-102 flower [38]. Li et al. proposed a text–image affine combination module (ACM) that could distinguish between the regions to be manipulated and those intended to be preserved [18,19]. Effective region selection using ACM suggests that the manipulation of more complex images, such as those contained in MSCOCO [39], is feasible. Moreover, a tremendous amount of research has been conducted on the manipulation of face images [21,22,23]. Face images can be manipulated via various manipulations, such as age, gender, facial expression, and hairstyle, in addition to color manipulation. These studies [21,22,23] were motivated by the capabilities of StyleGAN [40], which can produce realistic images in various domains. Moreover, with the emergence of a diffusion model [16], the accuracy of text-guided image manipulation has significantly improved [24,25]. The performance of generative models has recently improved dramatically, and it is expected that the performance of text-guided image manipulation will continue to improve in the future.

### 2.2. Image Quality Assessment

Research on image quality assessment has a long history and can be broadly classified into three categories—full-reference (FR), reduced reference, and no-reference—according to whether or not reference images are needed. In general, previous studies on text-guided image manipulation have evaluated the image generation performance of models by reconstructing the manipulated images from the input images. Since the input image can be used as a reference to evaluate the quality of the manipulated image, evaluation metrics based on FR are beneficial. In this section, we describe the evaluation metrics based on FR before and after the introduction of machine learning.

Before the introduction of machine learning into the field of image quality assessment, the mean squared error (MSE), peak signal-to-noise ratio (PSNR), and SSIM were typical FR-based evaluation metrics. MSE is computed by averaging the squared intensity differences between the pixels of the manipulated and the input images, and it is widely used because of its simplicity. PSNR is an evaluative metric that adjusts the value of MSE by using the maximum possible power of a signal and it can be considered as a type of MSE. MSE and PSNR fail to consider the content of an image and the characteristics of the human visual system (HVS), and their evaluation values usually show a weak correlation with human perception. As an evaluation approach that considers the HVS, Wang et al. proposed SSIM [33]. SSIM adopts the luminance comparison function based on Weber’s law, the contrast comparison function, and the structure comparison function, and the overall evaluation value computed from these three functions is designed to correspond with human perception.

Machine learning has proven to be successful in evaluating image quality. In particular, IS [34] and FID [35] are the most commonly used and have driven the rapid development of recent generative models. IS and FID extract high-level image features using the Inception-v3 model [41] and calculate an evaluation score based on these features. IS focuses specifically on the ease of identification by the Inception-v3 model and the variety of identified labels for the features of the generated images. IS uses only the image features of a generated image, whereas FID uses the image features of real and generated images and focuses on the similarities of these distributions. When IS and FID are applied to text-guided image manipulation, the real and generated images correspond to the input and manipulated images, respectively.

In text-guided image manipulation, manipulated images are evaluated from two perspectives: image quality and accuracy of image manipulation. These evaluation approaches are independent of each other and should be conducted separately. In this paper, we propose a new metric that evaluates the accuracy of image manipulation rather than image quality.

### 2.3. Assessment of Text-Guided Image Manipulation

The main purposes of text-guided image manipulation are to manipulate attributes in an image according to a text prompt and reconstruct the content in the text-irrelevant regions of the input image. In previous studies, the results were separately evaluated for these two objectives. In particular, one objective concerned the consistency between a manipulated image and a given text prompt, and the other concerned the similarity between a manipulated image and an input image. The details of these two evaluation approaches are described as follows.

The consistency between a manipulated image and a given text prompt is generally evaluated by focusing on the similarity between the two modalities. Particularly, previous studies, such as for the deep attentional multimodal similarity model [42] and the contrastive language-image pre-training (CLIP) [43], have embedded the manipulated image and text prompt in a common embedding space and then evaluated the models according to the cosine similarity between their embeddings. CLIP is pretrained on 400 million image–text pairs and is well known for its robust representation capabilities of images and texts in the common embedding space. Using CLIP, the extent to which a manipulated image reflects the content of a text prompt can be evaluated. Further, it is possible to derive Recall@*k* and R-precision from the calculated cosine similarity. Recent improvements in the common embedding space represented by CLIP hold promise for text-guided image manipulation evaluation; however, there is insufficient research on the approach to calculating evaluation scores based on such a space.

The similarity between the input and manipulated images is generally evaluated by focusing on the distance between them. Particularly, L1 and L2 distances have been used in previous research. With this calculation approach, distances increase even when the changes in a manipulated region are appropriate. The calculation of these distances is also sensitive to shifting positions and is difficult to generalize. The L1 or L2 distance is calculated by comparing an input image with a manipulated image at the pixel level, increasing these distances when there is a slight misalignment. However, in terms of human perception, a slight misalignment has only a minor effect on the accuracy of image manipulation.

These similarities and distances are usually calculated and evaluated separately. Moreover, Li et al. proposed MP [18], which uses “the cosine similarity between a manipulated image and text prompt” and “the L1 distance between input and manipulated images” and evaluates the accuracy of image manipulation that uses the text prompt. The similarity and the distance employed in MP indirectly evaluate the performance of text-guided image manipulation, and even if they are combined, the trade-off relationship may not be resolved.

In this study, we develop a new metric for evaluating the performance of text-guided image manipulation, categorized as the metric described in this section. Existing evaluation metrics separately evaluate image manipulation according to a text prompt and the preservation of text-irrelevant regions. Further, two evaluation scores calculated at different scales are forcibly merged. Moreover, the proposed metric, MD, focuses on text and image changes that occur before and after image manipulation and can comprehensively evaluate the performance of text-guided image manipulation.

## 3. MD-Based Evaluation of Text-Guided Image Manipulation

To evaluate the performance of text-guided image manipulation, we examine the case shown in Figure 1. In this figure, text changes are made by replacing a word related to an image attribute that a user wants to manipulate. We calculate the similarity between the *change in image features from the input to manipulated images* and the *change in text features by the word replacement for performing the manipulation* in the common embedding space. Thus, we name the evaluation metric “MD” because it is designed to focus on the direction of changes in images and texts that occur before and after image manipulation in the common embedding space. To perform the evaluation, as shown in Figure 3, we implement a two-step process: (1) the embedding of images and text prompts and (2) the calculation of the MD score. In step (1), we obtain four features of the input, manipulated images, and text prompts before and after the word replacement. Then, step (2) calculates the MD score using the four features based on the cosine similarity. We describe the two-step process in the following sections, Section 3.1 and Section 3.2, respectively.

### 3.1. Embedding of Image and Text Prompt

To perceive how changes occurred between image and text modalities based on image manipulation, we embed the input, manipulated images, and text prompts before and after the word replacement. To embed these modalities, we apply the recently proposed CLIP [43]. CLIP includes two neural networks of an image encoder $F_{\mathrm{image}}$ (·) and a text encoder $F_{\mathrm{text}}$ (·) and can extract effective feature vectors for matching images and texts. Particularly, the image encoder $F_{\mathrm{image}}$ (·) calculates the image features $x=F_{\mathrm{image}}$ (*i*) $\in R^{D}$ from an image *i*, and the text encoder $F_{\mathrm{text}}$ (·) calculates the text features $y=F_{\mathrm{text}}$ (*t*) $\in R^{D}$ from a text prompt *t*. Notably, *D* denotes the dimensionality of the common embedding space. During the training of the image and text encoders, CLIP maximizes the cosine similarity of *N* pairs of correct image and text embeddings in a batch and minimizes the cosine similarity of $N^{2}$–*N* pairs of incorrect embeddings. Finally, the image and text encoders are trained simultaneously to obtain a multimodal embedding space. Particularly, CLIP is trained on a very large image–text pair dataset and thus provides high expressive power for embedding.

### 3.2. Calculation of Evaluation Score via Proposed MD

We evaluate the performance of text-guided image manipulation using the elements that are processed and generated during the manipulation (i.e., the input image *I*, manipulated image $I^{\prime }$, original text prompt *T*, and replaced text prompt $T^{\prime }$). To calculate the MD score, we obtain the image and text features, $x_{I}=F_{\mathrm{image}}$ (*I*), $x_{I^{\prime }}=F_{\mathrm{image}}$ ($I^{\prime }$), $y_{T}=F_{\mathrm{text}}$ (*T*), and $y_{T^{\prime }}=F_{\mathrm{text}}$ ($T^{\prime }$) that exist in the common embedding space of CLIP. Next, we calculate the cosine similarity between “the change in the image features from $x_{I}$ to $x_{I^{\prime }}$” and “the change in the text features from $y_{T}$ to $y_{T^{\prime }}$” as follows:(1)$$\begin{array}{ccc} \mathrm{MD}(I,I^{\prime },T,T^{\prime }) & = & \frac{d_{I\to I^{\prime }}\cdot d_{T\to T^{\prime }}}{\left|\right|d_{I\to I^{\prime }}{\left|\right|}_{2}\left|\right|d_{T\to T^{\prime }}{\left|\right|}_{2}}, \end{array}$$(2)$$\begin{array}{ccc} d_{I\to I^{\prime }} & = & x_{I^{\prime }}-x_{I}, \end{array}$$(3)$$\begin{array}{ccc} d_{T\to T^{\prime }} & = & y_{T^{\prime }}-y_{T}, \end{array}$$
where $d_{I\to I^{\prime }}$ indicates the change from the input image *I* to the manipulated image $I^{\prime }$, and $d_{T\to T^{\prime }}$ indicates the change from the original text prompt *T* to the replaced text prompt $T^{\prime }$. Here, the MD score is calculated by using the cosine similarity between $d_{I\to I^{\prime }}$ and $d_{T\to T^{\prime }}$, and it takes a value between −1 and 1. A higher evaluation score of MD denotes a better text-guided image manipulation performance.

MD simultaneously considers, using the same scale, the image manipulation according to the text prompts and the preservation of text-irrelevant regions. This idea stems from the fact that $d_{I\to I^{\prime }}$ is naturally less similar to $d_{T\to T^{\prime }}$ when the accuracy of the image manipulation according to the text prompts, the preservation of text-irrelevant regions, or both, is low. Conversely, when the image changes accurately reflect the changes in the text prompt, the similarity between the two vectors increase. MD collaboratively evaluates how the image has changed before and after the image manipulation and whether this agrees with the text prompts.

MD also offers the advantage of being insensitive to errors caused by differences in the information content of images and texts. In text-guided image manipulation, the text prompt $T^{\prime }$ describes only where and how to manipulate an image. A direct comparison between the manipulated image $I^{\prime }$ and the text prompt $T^{\prime }$ is insufficient because it does not consider the information content of text-irrelevant regions in the image. As shown in Figure 3, the vectors $d_{I\to I^{\prime }}$ and $d_{T\to T^{\prime }}$ capture only the changes related to image manipulation in the image and text modalities, respectively. An evaluation based on the similarity between these vectors is unaffected by this difference in information content, which is our discovery.

## 4. Experiments

In this section, we validate the effectiveness of MD through extensive experiments. First, we verify its relevance to the subjective scores on the manipulated images in Section 4.1. We use the calculated MD score as a loss function and show the correspondence between the result of the image manipulation and the evaluation score in Section 4.2. Finally, the relationship between the common embedding space and the calculated MD score is presented in Section 4.3.

### 4.1. Verification of Correlation between Metrics and Subjective Evaluations

#### 4.1.1. Experimental Conditions

In MD, we applied CLIP [43], which can embed images and text prompts into a common embedding space. The image and text encoders for CLIP were ResNet101 [44] and transformer [2], respectively, and the embedding dimensionality *D* is set to 512. We used two datasets to evaluate the performance on text-guided image manipulation. First, we used Multi-Modal-CelebA-HQ [21], containing high-resolution face images of CelebA-HQ [45] and text prompts, to represent the images. Multi-Modal-CelebA-HQ has 30,000 images, each accompanied by 10 text prompts, and the images are split into 24,000 training images and 6000 test images. To evaluate image manipulation on Multi-Modal-CelebA-HQ, we used the text-guided image manipulation methods TediGAN [21], StyleMC [23], StyleCLIP-latent optimization (LO), and StyleCLIP-global direction (GD) [22]. These methods adopt StyleGAN [40] pretrained on FlickrFaces-HQ (FFHQ) [40] as their base architecture. Next, we adopted Caltech-UCSD Birds (CUB) [32]. CUB has 11,788 images, each accompanied by 10 text prompts, and the images are split into 8855 training images and 2933 test images. To evaluate image manipulation performance on CUB, we used the text-guided image manipulation methods TAGAN [17], ManiGAN [18], Li’20 [19], and Haruyama’21 [20], all pretrained on CUB. To obtain the manipulated images to be evaluated, we first randomly selected 120 pairs of images and text prompts from each dataset. Next, we created text prompts to be used for image manipulation by changing some of the words in the selected text prompts. Finally, each text-guided image manipulation method took the images and the changed text prompts as inputs and produces the manipulated images for evaluation.

To validate the effectiveness of MD, we employed cosine similarity (CS) and L1 distance (L1) as the comparative metrics, as well as manipulative precision (MP) [18] to evaluate the accuracy of the image manipulation. Generally, manipulated images are evaluated from two perspectives: image quality and the accuracy of image manipulation in text-guided image manipulation research. These evaluation approaches are independent of each other and must be conducted separately. In our experiment, we used the evaluation metrics for the accuracy of image manipulation as the comparison and we did not refer to the evaluation approaches for image quality, such as SSIM [33], IS [34], and FID [35]. As the comparative metrics, we adopted the evaluation metrics for the accuracy of image manipulation. The details are shown as follows.

CS calculates the cosine similarity between the manipulated image $I^{\prime }$ and the replaced text prompt $T^{\prime }$ in the common embedding space based on CLIP [43]. This metric takes a value between −1 and 1.L1 calculates the L1 distance between the input image *I* and the manipulated image $I^{\prime }$. This distance evaluates the image reconstruction performance, so a smaller score indicates better performance. To make it consistent with other metrics, the L1 distance is adjusted as follows:
(4)$$L1=1-\frac{1}{255\times CHW}\left|\right|I^{\prime }-I{\left|\right|}_{F},$$
where *C*, *H*, and *W* denote the number of color channels, height, and width of the images, respectively. This metric takes a value between 0 and 1.MP evaluates the performance of text-guided image manipulation by multiplying the normalized CS and L1 values. This metric takes a value between 0 and 1.

Notably, higher scores calculated by the above metrics indicate better image manipulation.

#### 4.1.2. Correlation between Metrics and Subjective Evaluations

We calculated a mean opinion score (the MOS) according to [33] for each manipulated image. To calculate the MOS, we first asked 28 subjects to evaluate the accuracy of 480 (4 image manipulation methods × 120 samples) manipulated images from each dataset on a five-point scale. Next, the raw scores for each subject were normalized using the mean and variance of all the scores given by its corresponding subject. Then, the normalized scores were rescaled so that all the scores would fall within the range of 1 to 100. Notably, the resizing process was performed for each dataset. Finally, we calculated the average score for each manipulated image as the MOS.

In Figure 4, we show each manipulated image generated on Multi-Modal-CelebA-HQ and its corresponding evaluation scores based on the proposed and comparative metrics. In this sample, the image manipulation performance is polarized between TediGAN and StyleCLIP-LO and StyleCLIP-GD and StyleMC. The former methods failed to reflect black hair, whereas the latter succeeded, and their MOS was in alignment with this. The evaluation scores by comparative metrics were limited in capturing this polarization. Moreover, MD tended to give high evaluation scores only to the manipulated images that contained the attribute of black hair. It is evident that the rank order of the MD score is well matched to that of the MOS. Furthermore, Figure 5 shows each manipulated image generated on CUB and its corresponding evaluation scores based on the proposed and comparative metrics. MD gives the highest score to the manipulated image generated from TAGAN, which is most successful in manipulating the image along the text prompt and preserving the text-irrelevant regions. For the results on CUB, the rank order of the MD score is also well matched to that of the MOS.

To verify the correlation between the MOS and the evaluation scores calculated by the proposed and comparative metrics, scatter plots are shown in Figure 6 and Figure 7 that correspond to the results based on Multi-Modal-CelebA-HQ and CUB, respectively. In these scatter plots, we indicate the linear functions based on linear regression analysis. In the validation results for both datasets, there appears to be no correlation between the MOS and the evaluation scores calculated by the comparative metrics. Specifically, CS has the characteristic of being insensitive to MOS changes, a subjective evaluation by humans. Some linear functions based on CS show a positive slope, but they are unreliable because the plots do not follow the line and are locally distributed. The evaluation scores based on L1 are sensitive to changes in the MOS. In general, the L1 distance is widely known to be uncorrelated with subjective evaluation and this might have caused the above results. The linear functions based on L1 have either near-zero or negative slopes. Since MP is calculated by multiplying CS and L1, it is obvious that MP is also uncorrelated, which can be confirmed from the scatter plots. Meanwhile, we confirmed that there was a correlation between the MD score and the MOS. The linear functions show positive slopes for all text-guided image manipulation methods applied to each dataset. The percentage of plots distributed along the functions is high and the slopes of the linear functions are more reliable than those of CS.

To quantitatively verify the relationship between the MOS and the evaluation score by each metric, following [33,46], we used the following two correlation coefficients and a distance measure: Pearson correlation coefficient (PCC), Spearman’s rank correlation coefficient (SRCC), and Earth mover’s distance (EMD) [47]. PCC and SRCC are well-known correlation coefficients, whereas EMD measures the proximity in the distributions of the MOS and the calculated evaluation scores. The results of PCC, SRCC, and EMD are shown in Table 1. From this table, it can be confirmed that MD outperforms the comparative metrics in terms of proximity of distribution and correlation coefficients with the MOS. Particularly, the values of PCC and SRCC between the MOS and each comparative metric do not exceed 0.2 and are generally low. Meanwhile, the values of PCC and SRCC based on MD are 0.547 and 0.490, respectively, on Multi-Modal-CelebA-HQ, and 0.316 and 0.323, respectively, on CUB. These results demonstrate that MD can calculate the evaluation scores that are positively correlated with the MOS compared with the comparative metrics. Further, in terms of EMD, the calculated MD score distribution is the closest to the MOS distribution. The above results indicate that MD is capable of calculating the evaluation score that is closest to the subjective scores for the manipulated image.

### 4.2. Verification of Correspondence between Results of Image Manipulation and MD Scores

In the above validation, because the image manipulation performance depends on existing text-guided image manipulation methods, there is a limitation in evaluating the extensive performance of image manipulation. This section verifies how the evaluation score calculated by the metrics changes in response to continuous changes in a manipulated image. We applied an evaluation metric to the loss function of the image manipulation network to qualitatively confirm the change in the evaluation score during training and the corresponding manipulated image.

#### 4.2.1. Experimental Conditions

Figure 8 shows an overview of the image manipulation network applying an evaluation metric to a loss function. In Figure 8, we employ StyleGAN [40] pretrained on FlickrFaces-HQ (FFHQ) [40] as the base architecture and optimize parameters inputting to it by maximizing the evaluation score computed via an evaluation metric. To implement text-guided image manipulation, we used the images and text prompts contained in Multi-Modal-CelebA-HQ [21] and changed the words in the text prompts. The changes in the text prompts correspond to what is shown on either side of the arrows in Figure 8. We optimized the parameters over 50 iterations for each image and obtained the manipulated image. To validate the trend in the evaluation scores for the manipulated images, we introduced CS, L1, MP, and MD as the loss function.

#### 4.2.2. Results of Image Manipulation Introducing MD as the Loss Function

Figure 9 shows the results of the image manipulation when MD is used as the loss function. This figure includes three results, with one each at 5, 20, and 50 iterations in each sample, to confirm how the manipulated images change corresponding to the change in the MD score. It can be visually confirmed that there is no discrepancy between the image manipulation accuracy and the corresponding evaluation score. Particularly, at five iterations (the orange points), none of the manipulated images in any of the samples reflect the text prompts and the corresponding MD scores are low. The results at 20 iterations (the yellow points) show that we successfully manipulated the images according to the text prompts, but the preservation of text-irrelevant content is inadequate. Potentially, the MD score at this iteration is high because the images were manipulated according to the text prompts. Finally, the manipulated images at 50 iterations (the blue points) show excellent results for all samples and the corresponding MD score is the highest of any number of iterations. The image content is manipulated according to the text prompt (e.g., no beard or glasses), whereas the input image content is maintained in text-irrelevant regions. These results suggest that MD is capable of coordinately evaluating image manipulation according to a text prompt and preservation of text-irrelevant regions on a single scale.

Figure 10 shows the comparative results of text-guided image manipulation where MD and comparative metrics are applied to a loss function. Notably, all the manipulated images are the results at 50 iterations and we have confirmed that the evaluation score by each metric was sufficiently high. When the comparative metrics are applied to the loss function, the accuracy of the text-guided image manipulation being inadequate can be verified. These results confirm the low association between the accuracy of image manipulation and the evaluation score calculated using comparative metrics. Meanwhile, when MD is applied to the loss function, it is established that a higher evaluation score indicates high accuracy in image manipulation.

### 4.3. Verification of Performance Dependence on Common Embedding Space

Finally, we verify the influence of the common embedding space used to calculate the MD score. Although we used the common embedding space provided by CLIP, it was confirmed in the previous verification that MD does not completely rely on its expressive capabilities. Particularly, CLIP is also applied to the calculation of CS and MP used as the comparative methods, and MD has a significantly higher correlation with the MOS than CS and MP. However, a strong embedding model such as CLIP is necessary to a highly accurate evaluation, and its expressive capabilities can influence MD. In the following, we verify the correlation between MD and the MOS using various common embedding spaces.

#### 4.3.1. Experimental Conditions

To embed images and text prompts into a common embedding space, we used CLIP [43], whose image encoder is ResNet [44] (ResNet50 or ResNet101) or vision transformer [3] (ViT-B/32 or ViT-L/14). Moreover, an existing visual-semantic embedding (VSE) model [48] pretrained on the Flicker30K [49] dataset was used. The embedding dimensionality *D* for each common embedding space is shown in Table 2.

**Figure 9 sensors-23-09287-f009:**
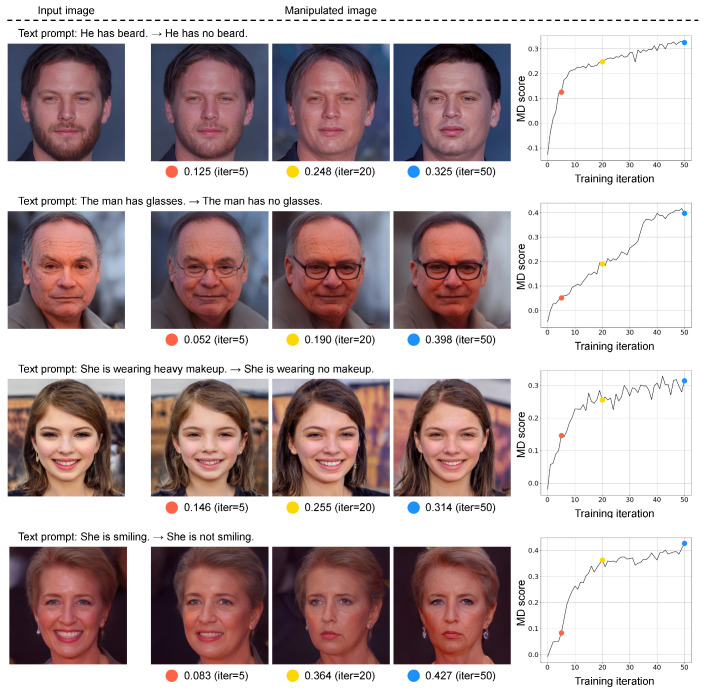
Results of image manipulation where MD is used as a loss function. The right-most column shows the progression of the calculated MD score over 50 iterations of parameter optimization. The orange, yellow, and blue dots represent the 5th, 20th, and 50th training points, respectively. We also show the generated images that correspond to these training points. Notably, the text prompts on either side of the arrows corresponds to *T* and $T^{\prime }$, respectively.

#### 4.3.2. Influence of Common Embedding Space on Performance of MD

We show the correlation coefficients between the MOS and the MD score calculated in different common embedding spaces in Table 2. Focusing on the difference in the evaluation performance between VSE and CLIP, the calculated MD score has high correlation with the MOS when CLIP is used. Since the common embedding space of CLIP is obtained based on a large number of image–text pairs, the space has robust expressive capabilities. Combining this capability with the evaluation approach of MD provides highly accurate evaluations for text-guided image manipulation. Further, even within CLIP, differences in the image encoder can affect the evaluation performance. ViT incorporates more global information than ResNet does at shallow layers and produces global features [50]. Whether local or global features are suitable for calculating MD depends on the images in a dataset. Therefore, selecting a suitable encoder according to the evaluation images is necessary.

**Figure 10 sensors-23-09287-f010:**
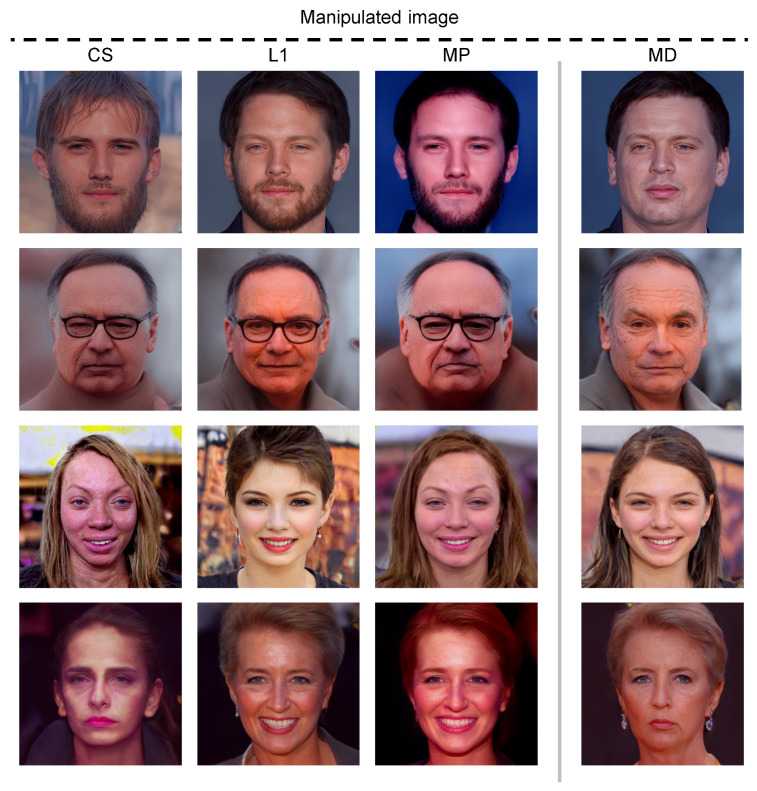
Visual comparison of image manipulations where MD and comparative metrics are used to loss functions. The numbers in parentheses are evaluation scores by each metric after 50 iterations of parameter optimization. The input images and text prompts for generating manipulated images are the same as those shown in Figure 9.

## 5. Conclusions

We proposed a logical and robust metric, MD, to evaluate the accuracy of text-guided image manipulation. MD calculates the similarity between *the change in the image modality* and *the change in the text modality* occurring before and after image manipulation. The evaluation using MD simultaneously considers both image manipulation according to text prompts and the preservation of text-irrelevant regions on the same scale. The experimental results demonstrated that MD could calculate evaluation scores more strongly correlated with the MOS than the existing metrics. Further, we constructed an image manipulation network applying evaluation metrics to a loss function and we confirmed that the manipulated images improved as the MD score increased. The approach of calculating MD to focus on the relationship between image and text changes is useful for evaluating the accuracy of text-guided image manipulation, but the performance of the embedding model is necessary for a highly accurate evaluation. Expectedly, the accuracy of MD will improve in the future as the common embedding space becomes more robust. In future works, to evaluate a variety of image manipulation, we plan to construct an evaluation metric that focuses not only on the global features of the whole image, but also on local features. 

## Figures and Tables

**Figure 1 sensors-23-09287-f001:**
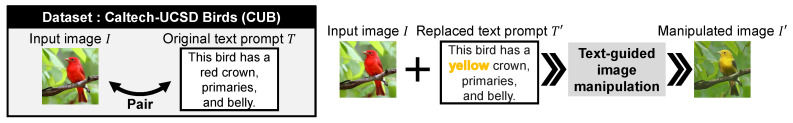
Example of text-guided image manipulation on Caltech-UCSD Birds (CUB) [32]. The yellow word in the text prompt $T^{\prime }$ is a replaced word and indicates manipulation.

**Figure 2 sensors-23-09287-f002:**
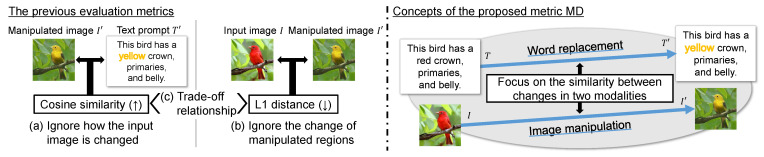
Top depicts specific cases of problems (a)–(c) with existing evaluation metrics. Bottom depicts the concept of the proposed metric, MD, that comprehensively evaluates the performance of text-guided image manipulation on a single scale, unlike the existing metrics.

**Figure 3 sensors-23-09287-f003:**
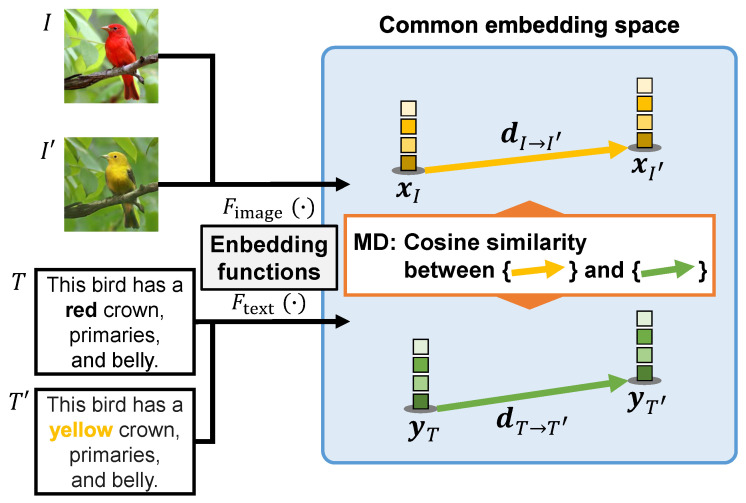
Overview of MD. Notably, $x_{I}$, $y_{T}$, $x_{I^{\prime }}$, and $y_{T^{\prime }}$ are image and text features in the common embedding space, calculated by applying the embedding functions $F_{\mathrm{image}}$ (·) and $F_{\mathrm{text}}$ (·), respectively.

**Figure 4 sensors-23-09287-f004:**
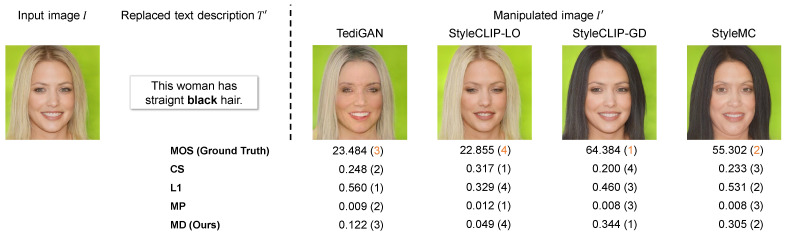
Sample of evaluation scores calculated by MD and comparative metrics for text-guided image manipulation on Multi-Modal-CelebA-HQ [45]. The bold word in the replaced text prompt $T^{\prime }$ was “blond” in the original text prompt *T*. The numbers in parentheses indicate the rank of the evaluation score based on the MOS or each evaluation metric. Note that the acceptable range of the MOS is between 0 and 100, according to [33], and its calculation is described in Section 4.1.2.

**Figure 5 sensors-23-09287-f005:**
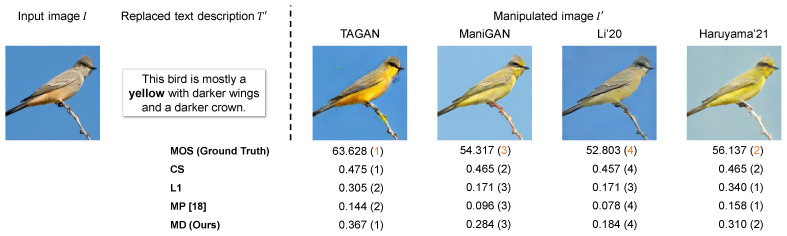
Sample of evaluation scores calculated by MD and comparative metrics for text-guided image manipulation on CUB [32]. The bold word in the replaced text prompt $T^{\prime }$ was “gray” in the original text prompt *T*. The numbers in parentheses indicate the rank of the evaluation score based on the MOS or each evaluation metric. Note that the acceptable range of the MOS is between 0 and 100, according to [33], and its calculation is described in Section 4.1.2.

**Figure 6 sensors-23-09287-f006:**
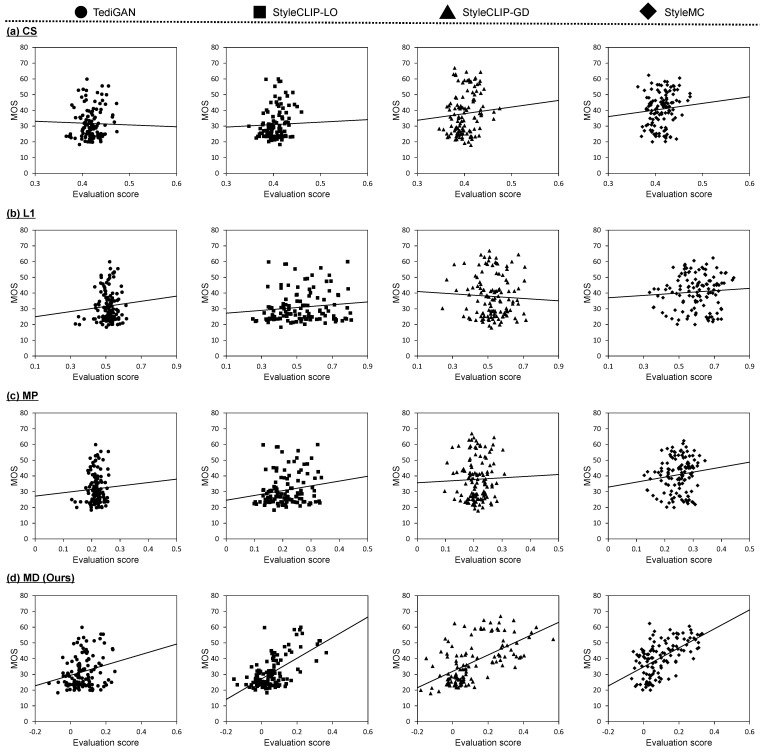
Scatter plots of the MOS versus evaluation scores calculated by each metric on Multi-Modal-CelebA-HQ. Each dot represents an evaluation of one manipulated image used in subject experiments. Results for (**a**) CS, (**b**) L1, (**c**) MP [18], and (**d**) MD are shown from top to bottom. The linear function shown in each scatter plot was obtained via linear regression analysis.

**Figure 7 sensors-23-09287-f007:**
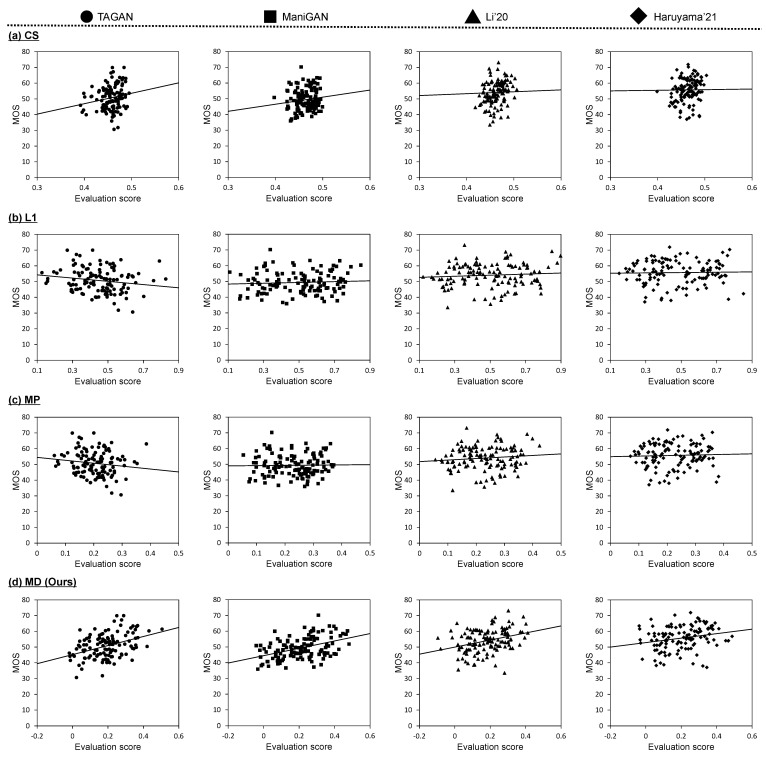
Scatter plots of the MOS versus evaluation scores calculated by each metric on CUB. Each dot represents an evaluation of one manipulated image used in subject experiments. Results for (**a**) CS, (**b**) L1, (**c**) MP [18], and (**d**) MD are shown from top to bottom. The linear function shown in each scatter plot was obtained via linear regression analysis.

**Figure 8 sensors-23-09287-f008:**
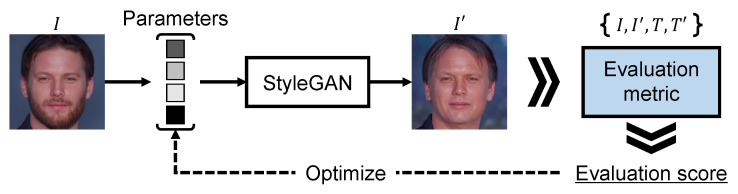
Overview of the image manipulation network applying the evaluation metric to a loss function. The manipulated image $I^{\prime }$ is obtained by optimizing the parameters inputting to StyleGAN according to the evaluation scores. Notably, the optimization is performed over 50 iterations.

**Table 1 sensors-23-09287-t001:** Results of PCC, SRCC, and EMD between the MOS and the evaluation score calculated by each metric on Multi-Modal-CelebA-HQ and CUB. Notably, marks (↑) and (↓) indicate that higher and lower are better. Bold values indicate the best results.

	Dataset: Multi-Modal-CelebA-HQ [21]			Dataset: CUB [32]		
	**PCC (↑)**	**SRCC (↑)**	**EMD (↓)**	**PCC (↑)**	**SRCC (↑)**	**EMD (↓)**
CS	0.178	0.159	0.136	0.120	0.197	0.096
L1	0.136	0.136	0.169	0.013	0.002	0.043
MP [18]	0.180	0.174	0.151	0.022	0.008	0.056
MD (Ours)	**0.547**	**0.490**	**0.095**	**0.316**	**0.323**	**0.029**

**Table 2 sensors-23-09287-t002:** Comparative results of PCC and SRCC between the MOS and MD scores for each common embedding space. Notably, the mark (↑) indicates that higher is better. Bold and underlined values indicate the first and second best results, respectively. The value of *D* indicates the embedding dimensionality.

		Multi-Modal-CelebA-HQ [21]		CUB [32]	
	D	**PCC (↑)**	**SRCC (↑)**	**PCC (↑)**	**SRCC (↑)**
MD with VSE [48]	300	0.205	0.205	0.232	0.241
MD with CLIP (ResNet50)	1024	0.545	**0.494**	0.287	0.300
MD with CLIP (ResNet101)	512	**0.547**	0.490	**0.316**	**0.323**
MD with CLIP (ViT-B/32)	512	0.444	0.424	0.312	0.310
MD with CLIP (ViT-L/14)	768	0.514	0.465	0.248	0.243

## Data Availability

A publicly available datasets were used in this work. The datasets can be found here: https://www.vision.caltech.edu/datasets/cub_200_2011/ and https://github.com/IIGROUP/MM-CelebA-HQ-Dataset (accessed on 14 November 2023).

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
