# Peer review of "Manipulation Direction: Evaluating Text-Guided Image Manipulation Based on Similarity between Changes in Image and Text Modalities"

_sensors, 2023, doi:10.3390/s23229287_

Round 1

Reviewer 1 Report

Comments and Suggestions for Authors

Comments on the Quality of English Language

Reviewer 2 Report

Comments and Suggestions for Authors

Dear Authors,

Appreciate your efforts in preparing manuscript titled "Manipulation Direction: Evaluating Text-guided Image Manipulation Based on Similarity Between Changes of Image and Text Modalities". Manuscript in the current version written well with clear research objective and evidence but there are few minor corrections would greatly benefits the manuscript in improving the quality and clarity. 

Reviewer 3 Report

Comments and Suggestions for Authors

This paper proposes a text-guided image manipulation method based the similarity between the changes of image and text. Experimental results show the proposed method outperforms the state-of-the-art approaches. There are some problems of theoretical and experimental analyses in this manuscript and it can be revised in the following aspects.

1. In introduction section, the main contributions of this paper should be clearly listed.

2. All mathematical equations should be carefully check to avoid possible mistakes. For example, in equation (1), the mathematical expressions of x_I, x_I’, y_I and y_I’ should be supplemented; in the title of table 2, ‘d’ should be ‘D’.

3. In figure 5, the comparison is based on single image. The comparison on a whole dataset is more suitable.

Comments on the Quality of English Language

The whole paper should be carefully checked to avoid some possible grammatical and typographical errors.

Round 2

Reviewer 1 Report

Comments and Suggestions for Authors

good paper

Comments on the Quality of English Language

good paper